# Acromial Morphology and Its Relation to the Glenoid Is Associated with Different Partial Rotator Cuff Tear Patterns

**DOI:** 10.3390/jcm12010233

**Published:** 2022-12-28

**Authors:** Paul Borbas, Rebecca Hartmann, Christine Ehrmann, Lukas Ernstbrunner, Karl Wieser, Samy Bouaicha

**Affiliations:** 1Department of Orthopaedic Surgery, Balgrist University Hospital, 8008 Zurich, Switzerland; 2RNR Radiologie und Neuroradiologie am Glattzentrum, 8304 Wallisellen, Switzerland

**Keywords:** acromion morphology, acromioplasty, PASTA lesion, partial supraspinatus tendon tears

## Abstract

The pathogenesis of subacromial impingement syndrome is controversially discussed. Assuming that bursal sided partial tears of the supraspinatus tendons are rather the result of a direct subacromial impact, the question arises whether there is a morphological risk configuration of the acromion and its spatial relation to the glenoid. Patients who underwent arthroscopic repair of either a partial articular supraspinatus tendon avulsion (PASTA) or bursal-sided supraspinatus tear (BURSA) were retrospectively allocated to two groups. Various previously described and new omometric parameters on standard anteroposterior and axial shoulder radiographs were analyzed. We hypothesized that acromial shape and its spatial relation to the glenoid may predispose to a specific partial supraspinatus tendon tear pattern. The measurements included the critical shoulder angle (CSA), the acromion index (AI), Bigliani acromial type and the new short sclerotic line, acromioclavicular offset angle (ACOA), and AC offset. The ratio length/width of acromion and the medial acromial offset were measured on axial radiographs. A total of 73 patients were allocated to either PASTA (*n* = 45) or BURSA (*n* = 28). The short sclerotic line showed a statistically significant difference between PASTA and BURSA (16.2 mm versus 13.1 mm, *p* = 0.008). The ratio acromial width/length was statistically significant (*p* = 0.021), with BURSA having slightly greater acromial length (59 vs. 56 mm). The mean acromial offset was 42.9 mm for BURSA vs. 37.7 mm for PASTA (*p* = 0.021). ACOA and AC offset were both higher for BURSA, without reaching statistical significance. The CSA did not differ significantly between PASTA and BURSA (33.73° vs. 34.56°, *p* = 0.062). The results revealed an association between a narrow acromial morphology, increased medial offset of the acromion in relation to the glenoid, and the presence of a short sclerotic line in the anteroposterior radiograph in bursal-sided tears of the supraspinatus tendon. Assuming that bursal-sided tears are rather the result of a direct conflict of the tendon with the undersurface of the acromion, this small subgroup of patients presenting with impingement syndrome might benefit from removing a harming acromial spur.

## 1. Introduction

In 1972, Neer postulated that certain acromial morphology can lead to direct impairment and consecutive tearing of the rotator cuff [1]. Later on, this concept was referred to as the extrinsic theory of rotator cuff degeneration [2]. Due to this “mechanical” impingement of the acromial roof with the rotator cuff tendons, Neer proposed concomitant acromioplasty during surgical repair of rotator cuff tears in order to avoid further mechanical irritation of the repaired tendon.

In contrast to Neer’s theory, the intrinsic theory of rotator cuff pathology was proposed. This theory stated that lesions of the rotator cuff occur secondary to intratendinous degeneration or tendinosis [3,4,5]. Rotator cuff tears are, therefore, assumed to be caused by eccentric tensile overload at a rate exceeding the ability of the cuff to repair itself [3,5]. According to this theory, acromioplasty, therefore, does not address the problem of tendon degeneration [3].

Recent studies suggest that the tensile overload of the supraspinatus tendon is correlated with scapular geometry [6]. Anatomical factors, such as the critical shoulder angle (CSA) or the lateral extension of the acromion (acromion index), are thought to influence biomechanical properties of glenohumeral force distribution during motion, thereby playing a role in development of rotator cuff tears [7,8]. In the literature, a larger CSA of 38° is associated with a higher prevalence of supraspinatus tears compared to disease-free shoulders with a CSA of 33° [9]. Biomechanical research supports the hypothesis that a larger CSA increases the ratio of joint shear to joint compression forces, thus leading to a substantially increased compensatory supraspinatus load that could lead to a rotator cuff tear [10]. These findings are widely contradictory to Neer’s idea of the pathogenesis of rotator cuff tendon tears.

However, anterior acromioplasty is still used for subacromial impingement that is refractory to nonoperative treatment and as an adjunct during arthroscopic rotator cuff repair [2]. Although several studies showed good outcomes following rotator cuff repair with acromioplasty [11,12,13], more recent investigations questioned the benefit of acromioplasty as an adjunct to rotator cuff repair, as the clinical results with and without acromioplasty were similar [14,15].

The benefit of an isolated arthroscopic acromioplasty in subacromial impingement syndrome is questioned in the literature, such as the CSAW trial, which showed no benefit of subacromial decompression over investigational arthroscopy only [16]. Neither subacromial decompression nor arthroscopy led to a clinically relevant improvement in patient reported outcomes when compared to no treatment. Even though the numbers of this surgical procedure rapidly declined over the past years, clinical experience nevertheless shows that a subgroup of patients can still benefit from this intervention in the longer term.

There is some evidence describing differences in the pathogenesis of bursal- versus articular-sided partial tears of the supraspinatus tendon, raising the question of the existence of a morphological risk configuration of the acromial shape and its spatial position, which is associated with bursal- but not articular-sided partial tears [5,17,18].

To our knowledge, up to date, no study on acromial morphology with regard to different supraspinatus tendon tear patterns has been conducted. It was, therefore, the aim of the present study to analyze dedicated omometric parameters in patients with either articular-sided or bursal-sided partial supraspinatus tendon tears. The hypothesis was that acromial shape and its spatial relation to the glenoid may predispose to a specific partial supraspinatus tendon tear.

## 2. Materials and Methods

The study was approved by the local ethical committee. 

We retrospectively identified all patients who underwent an arthroscopic repair of either an articular- or bursal-sided partial supraspinatus tear in our institution between 2011 and 2017. Inclusion criteria consisted of a complete medical history with available pre- and postoperative true anteroposterior and axial shoulder radiographs and preoperative shoulder MR arthrography.

Patients with an acromio-humeral distance of <7 mm, and hence, possible wear of the superior glenoid rim, inflammatory disease, previous surgery, and incomplete radiological imaging, were excluded. 

Patients were either allocated to the cohort with a partial articular supraspinatus tendon avulsion (PASTA) or bursal-sided (BURSA) supraspinatus tear accordingly, and various omometric parameters on standard anteroposterior and axial shoulder radiographs were analyzed. Omometry is a previously described umbrella term that contains all standardized measurements on plain radiographs of the shoulder, analogous to the term “coxometry”, which is widely used for the radiographic assessment of the osseous pelvis and hip [19].

All measurements were taken electronically on radiographs displayed on a PACS workstation (Cerner Corp., Kansas City, MI, USA). The omometric parameters were measured by two independent readers—a MSK trained radiologist (C.E.) and an orthopedic surgeon (R.H.). The average values of both measurements were used for final analysis. 

Operative reports were compared to the preoperative shoulder MR arthrographies.

### 2.1. Radiographic Measurements

On the AP shoulder radiographs, the CSA and acromion index were measured [8]. 

A new parameter, the short sclerotic line, was defined by the visible sclerotic line formed by the width of the acromion. The distance was measured on anteroposterior shoulder radiographs in mm (Figure 1 and Figure 2). 

The acromioclavicular offset angle (ACOA) and the AC offset (in mm) were also measured as new additional parameters on anteroposterior radiographs (Figure 3). The ACOA is an angle similar to the CSA, but measured between a line connecting the inferior with the superior border of the glenoid fossa and another connecting the inferior border of the glenoid with the most inferolateral point of the distal clavicle at the level of the AC joint. The AC offset is the distance measured perpendicular from the glenoid line (as of the CSA and ACOA) to the most inferolateral point of the distal clavicle.

On the axial radiographs, the length and width of the acromion and its ratio were measured (Figure 4 and Figure 5, bright yellow lines). In addition, the distance between a line at a square angle from the center of the glenoid to the tangent on the medial side of the acromion was measured. This parameter was called medial acromial offset (Figure 4 and Figure 5, red line). Acromial type was assessed according to Bigliani et al., and divided into either type 1 (flat), type 2 (curved), or type 3 (hook) [20].

### 2.2. Reliability of Radiographic Measurements

The impact of vertical and horizontal malrotation on measurements of anteroposterior radiographs of the scapula has already been studied previously. The measured values on the anteroposterior radiograph in this study remained stable within a narrow range of +/−9° of malrotation. Beyond this range, values of measured parameters significantly deviated [19]. 

The reliability depending of the scapular rotation of our newly defined parameters is unknown. With an in-study investigation, we therefore assessed the new parameters in vitro with a dry bone human scapula cadaveric specimen. We incrementally rotated (steps of 5°) the scapula around its transversal (flex/ex) and sagittal (IR/AR) axis, with a total range of 30° per each axis. The three newly defined parameters were measured with every incremental rotation and the change was noted. 

In the transversal plane, the medial border offset remained stable within 5° of malrotation with 2.7% of change in extension and 4.9% of change in flexion malrotation. The ratio of width/length of the acromion remained stable up to 10° of malrotation in each direction, with a value of 0.59 in neutral rotation and 0.58 with 10° of flexion malrotation and 0.61 with 10° of extension malrotation (Table 1). 

In the sagittal plane, the medial border offset remained fairly stable within 5° of abduction malrotation, with 9% of change, but showed greater deviation for 5° of adduction malrotation, with a change of 11%. The ratio width/length remained stable up to 10° of malrotation, with a value of 0.45 in the neutral position and 0.47 for 10° of adduction and 0.49 for 10° of abduction malrotation (Table 1).

In both planes, measured values remained stable within 5° of malrotation in each direction, but showed greater deviation with a malrotation of 10° and more. The new parameters should, therefore, only be measured on correct axial shoulder X-rays, which was the case for all images in our study cohort. 

### 2.3. Data Processing and Statistical Analysis

Study data were collected and managed using REDCap^®^ (Version 6.7.4, Vanderbilt University, Nashville, TN, USA; https://www.project-redcap.org). 

Statistical analyses were performed using IBM SPSS Statistics version 19. An independent samples *t*-test was performed for the measured parameters. A ROC curve and odds ratio was calculated for the short sclerotic line. Any *p*-value < 0.05 was considered to be statistically significant.

## 3. Results

Age in both groups was comparable with a mean age of 53 (range, 26–75) years for the PASTA and a mean age of 57 (range, 29–73) years for the BURSA group (*p* = 0.95). Due to the small sample size, matching of age and sex was not possible (Table 2). 

The PASTA group was comprised of a consecutive series of 46 patients (25 male, 21 female) who underwent unilateral arthroscopic repair of the rotator cuff for an articular-sided partial-thickness tear of the supraspinatus tendon. 

The BURSA group consisted of a consecutive series of 28 patients (17 male, 11 female) who underwent unilateral arthroscopic repair of the rotator cuff for a bursal-sided partial-thickness tear of the supraspinatus tendon. 

Results are summarized in Table 2. On the anteroposterior radiographs, the mean CSA was 33.73° (range, 24.85–44.08) in the PASTA and 34.56° (range, 28.9–44.29) in the BURSA group, not being statistically significant (*p* = 0.062).

The mean acromion index was 0.72 (range, 0.60–0.84) for the PASTA and 0.73 (range, 0.56–0.84) for the BURSA group. This difference was also insignificant (*p* = 0.44).

The difference in the short sclerotic line measured on the anteroposterior shoulder radiograph was highly significant (*p* = 0.008), with 16.2 mm (range, 6.64–28.26) in the PASTA compared to of 13.1 mm (range, 8.27–23.87) in the BURSA group.

The mean ratio between acromial width and length on the axial radiograph was 0.48 (range, 0.37–0.67) in the PASTA vs. 0.45 (range, 0.35–0.58) in the BURSA group. This difference was found to be statistically significant (*p* = 0.021), with the BURSA having a slightly greater acromial length than the PASTA group; 59 mm (range, 40.5–80.05) versus 56 mm (range, 38.15–73.78). The medial acromial border offset was significantly larger with 42.9 mm (range, 18.32–55.46) in the BURSA group versus 37.7 mm (range, 22.99–55.46) in the PASTA group (*p* = 0.021). 

The mean ACOA was 12.72° (range, 4.21–18.45) in the BURSA and 11.76° (range, 5.81–19.2) in the PASTA group, respectively (*p* = 0.349). The AC offset was again higher in the BURSA with 12.78 mm (range, 4.14–17.95) compared to 11.7 mm (range, 5–20.09) in the PASTA group, without reaching statistical significance (*p* = 0.294). 

For a short sclerotic line <12 mm, the odds ratio to belong to the BURSA instead of the PASTA group was 3.3 (*p* = 0.029), with a sensitivity of 77% and a specificity of 46% (Figure 6). 

There were 7 shoulders (25%) with Bigliani acromial type 1, 18 (64%) with type 2, and 3 shoulders (11%) with 3 in the BURSA, and 14 (30%) with type 1, 24 (52%) with type 2, and 8 shoulders (17%) in the PASTA group. The difference in acromion morphology according to Bigliani between both groups was not statistically significant (*p* = 0.562).

## 4. Discussion

The main finding of the present study is that we could verify our hypothesis of morphological risk factors of the scapula including several new radiographic parameters, which are associated with the presence of a bursal-sided partial tear of the supraspinatus tendon. 

The patients in the BURSA group had a larger medial acromial offset and the ratio of the acromial width/length was significantly smaller, i.e., the acromion was relatively narrow compared to the PASTA group. The typical appearance of this narrow acromion with a large offset to the glenoid was called “Ladyfinger” according to the intraoperative similarity to a ladyfinger biscuit after the acromioplasty was performed. Further, the sclerotic line measured on the anteroposterior radiograph represents a narrow acromion and was also significantly shorter in the BURSA group. 

Dietrich et al. found that a posterior acromial slope of more than 36° and the presence of calcific tendinitis on conventional shoulder radiography led to a significantly better clinical outcome in patients receiving fluoroscopy-guided subacromial injections [21]. The same study group also found a short lateral extension of the acromion, defined by a CSA < 35°, to be associated with better clinical outcomes in patients receiving subacromial injections, but not in glenohumeral injections [22]. These results possibly indicate that certain omometric parameters may influence clinical results of therapeutic shoulder interventions. Liu et al. concluded in a retrospective study of partial articular-sided supraspinatus tendon tears, that high CSA and acromion index are associated with these tears [23]. However, in the present study both the CSA and acromion index did not significantly differ between the PASTA and BURSA group. 

Ozaki et al. examined 200 shoulders from cadaveric specimens [18]. They found that in each specimen that had an incomplete bursal-sided supraspinatus tear, there was an attritional lesion on the coracoacromial ligament and also on the anterior one-third of the undersurface of the acromion. On the other hand, all specimens that had an incomplete articular-sided supraspinatus tear, the undersurface of the acromion was intact. In 1999, Sano et al. examined 76 cadaveric shoulders and concluded that intrinsic degeneration occurred foremost in the articular but not the bursal side of the rotator cuff [5]. This supports the findings of Fukuda et al., who examined histologic sections from 12 en bloc surgical specimens of patients with a bursal-sided partial rotator cuff tear [17]. In all 12 specimens, a subacromial tendon impingement could be confirmed intraoperatively. Panni et al. also noted that bursal-sided and complete cuff tears were associated with severe degenerative changes in the acromion in 100% of cases [24]. In their study, articular-sided cuff tears were not related either to acromial morphology or degenerative changes in the coracoacromial arch. They state that their results suggest that the incidence and severity of rotator cuff tears are correlated with aging and morphology of the acromion.

Recently, the sole mechanical genesis of rotator cuff tears has been doubted. There is evidence indicating that an additional acromioplasty does not improve clinical outcome after arthroscopic repair of the rotator cuff tear [13,14,25,26,27]. In most of these studies, patients with full-thickness tears were included and the shape of the acromion, as described by Bigliani et al., was assessed. However, only one study included patients with Bigliani Type-1 acromion as well; the other studies did not include Type-1 acromion and one study excluded Type-3 acromion. Our results confirm the findings of Pandey et al., that there is no correlation between the Bigliani acromial shape and the type of supraspinatus tear [28].

MacDonald showed a higher revision rate in patients with a Type-2 or Type-3 acromion who received only rotator cuff repair without acromioplasty [13]. Quality of life, patient-reported pain, and functional outcomes, however, did not differ significantly. 

Our findings may support, but not prove, the assumption that bursal-sided partial tears of the supraspinatus tendon are rather the result of a direct subacromial conflict than articular-sided tears, and therefore represent a specific subgroup of patients with subacromial pain. If this is the case, individuals suffering from a bursal-sided supraspinatus tear might be the few among the large group of patients presenting subacromial impingement syndrome who could potentially benefit from anterior acromioplasty, resulting in subacromial clearance and consecutively less friction between the subacromial undersurface and the rotator cuff according to Neer’s Theory. 

A possible explanation of such a local mechanical irritation could be the narrow and/or relatively to the glenoid localized more lateral point of “counter-pressure”, increasing the pressure on the poorly vascularized part of the supraspinatus tendon. We also measured the CSA and acromial index in both groups. No significant difference was found between the articular- and bursal-sided tears, indicating that the medial acromial offset is not just another way to measure lateral extension of the acromion, but rather an independent radiographic parameter. 

The position of the AC joint could also play a role in the development of a bursal-sided supraspinatus tear. Especially in case of AC joint arthritis; inferior osteophytic spurs may cause a mechanical conflict with the rotator cuff [29]. We have, therefore, also measured the lateral offset of the AC joint on anteroposterior radiographs by introducing two new omometric parameters, the AC offset angle and the AC offset. Both parameters were higher in the BURSA group, however, without reaching statistically significant difference in our patient population.

The present study has several limitations. Apart from the retrospective study design with considerably small numbers, the focus lies on radiological parameters only, not including clinical and patient-reported outcome measures. Further, the parameters measured on the anteroposterior and axial radiographs are sensitive to malrotation, as shown with our in-study investigation. Therefore, the newly introduced parameters may not be assessed on radiographs of low quality. However, when intentionally malrotating the scapula in our study, the “wrong” position of the X-ray beam was immediately obvious after inadequate images were obtained. In consequence, further (prospective) clinical investigations and biomechanical proof-of-concept studies are needed to identify a possible subgroup of patients who could benefit from anterior acromioplasty.

## 5. Conclusions

A narrow acromion with a large offset of its medial border with regard to the glenoid plane is associated with bursal-sided partial tears of the supraspinatus tendon. Assuming that this subgroup of tears is caused by direct mechanical irritation of the acromion undersurface and consecutively leads to wear of the underlying tendon, identification of the newly introduced radiographic parameters may be helpful in assessing patients with subacromial impingement syndrome. Further studies are needed to verify the potential benefit of anterior acromioplasty with or without rotator cuff repair in patients presenting the newly introduced morphological characteristics of the acromion. 

## Figures and Tables

**Figure 1 jcm-12-00233-f001:**
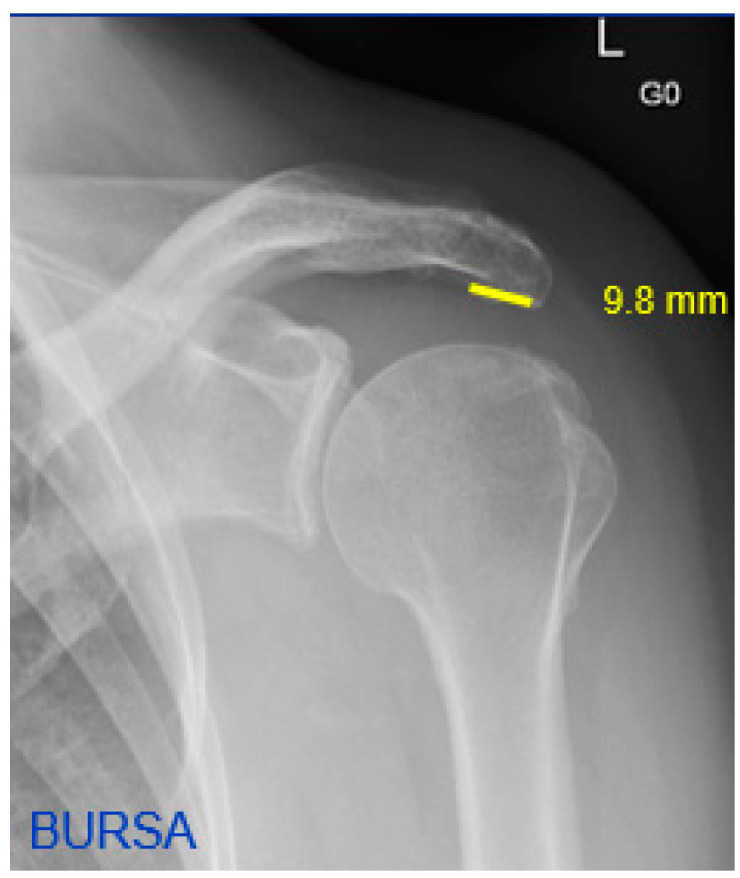
Left shoulder anteroposterior radiograph of the BURSA group indicating a “Short Sclerotic Line” (in yellow).

**Figure 2 jcm-12-00233-f002:**
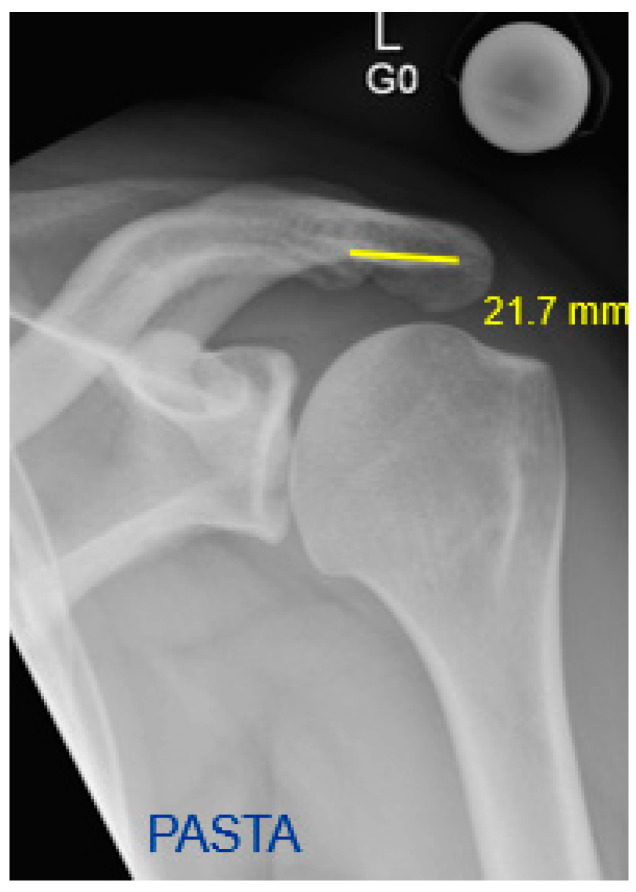
Left shoulder anteroposterior radiograph of the PASTA group indicating a longer sclerotic acromial line (in yellow).

**Figure 3 jcm-12-00233-f003:**
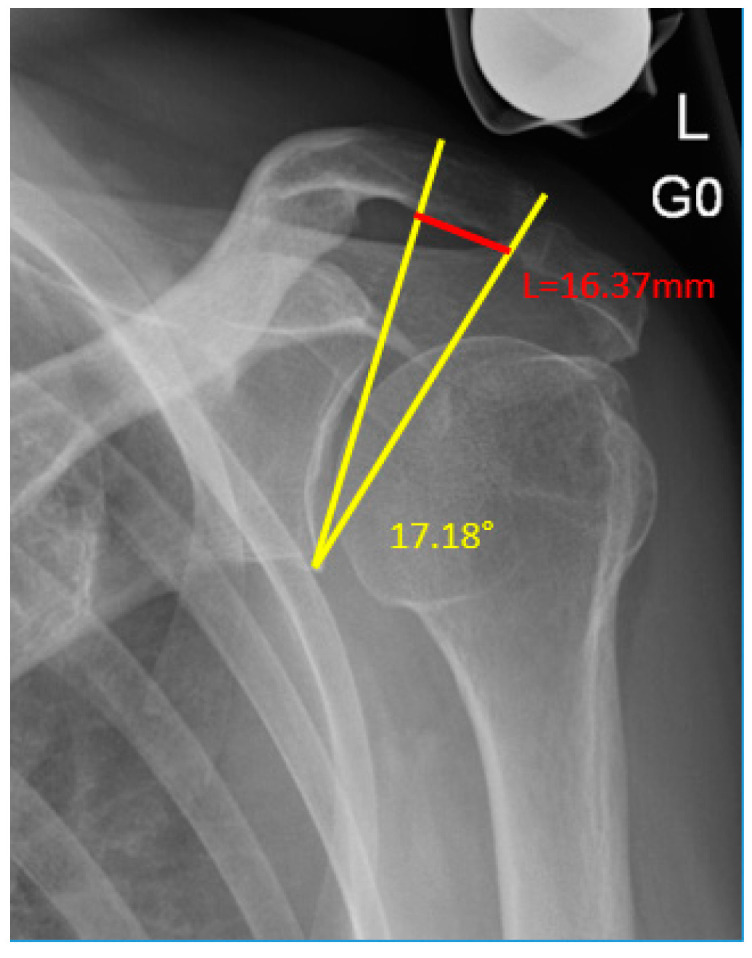
Left shoulder anteroposterior radiograph illustrating the acromioclavicular offset angle (ACOA, in yellow) and the AC offset (in red).

**Figure 4 jcm-12-00233-f004:**
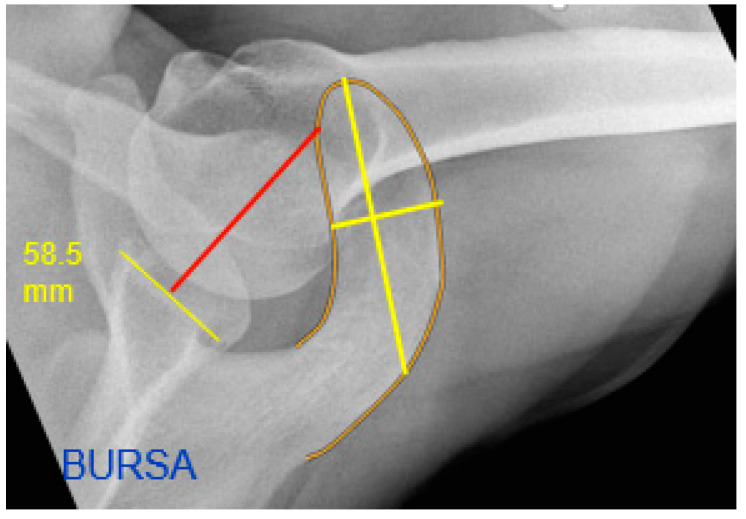
Left axillary shoulder radiograph: medial border offset (red), length/width of the acromion (bright yellow).

**Figure 5 jcm-12-00233-f005:**
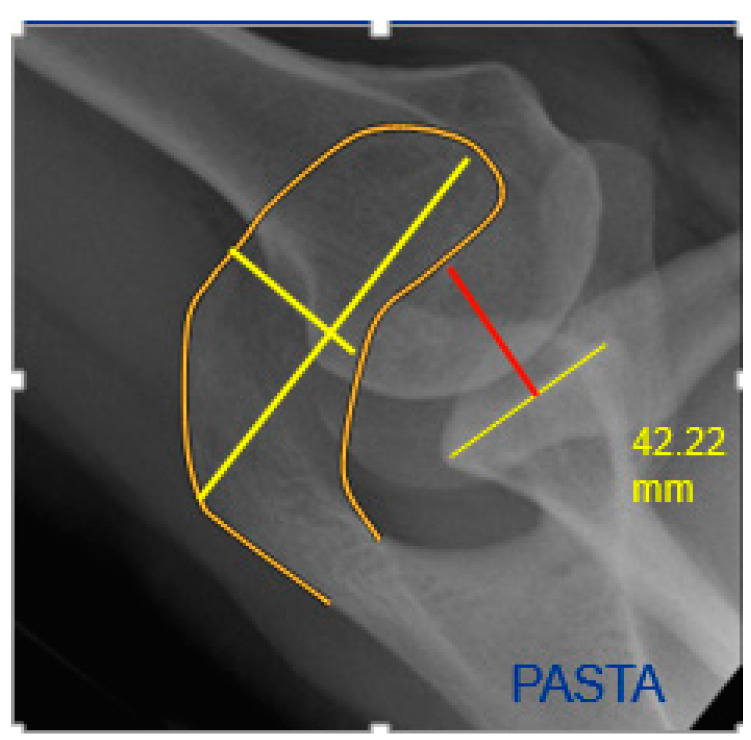
Right axillary shoulder radiograph: medial border offset (red), length/width of the acromion (bright yellow).

**Figure 6 jcm-12-00233-f006:**
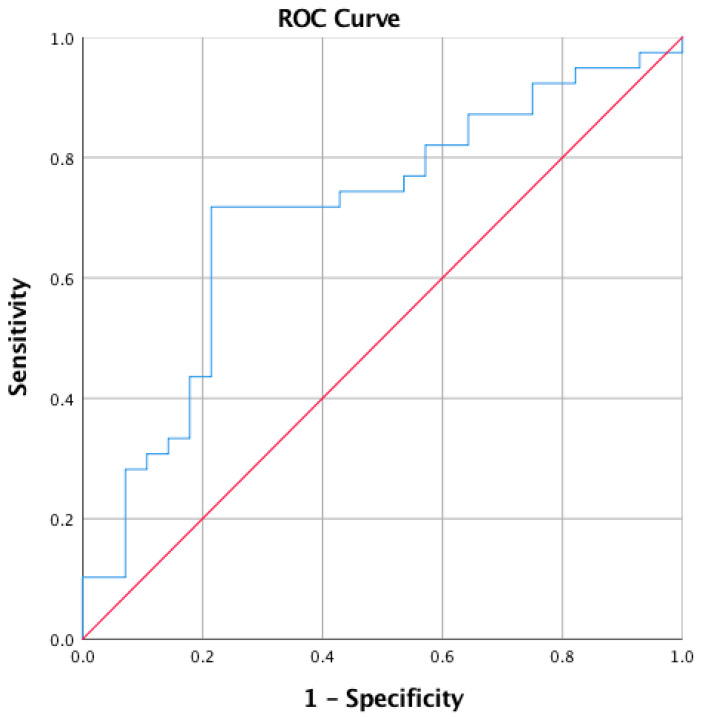
ROC curve for short sclerotic line <12 mm.

**Table 1 jcm-12-00233-t001:** Malrotation influence.

Position	Medial Offset (in mm)	% Change	Ratio Width/Length
Neutral	223	0	0.59
5° Flex	212	−4.9	0.57
10° Flex	184	−17.5	0.58
15° Flex	144	−35.4	0.55
5° Ext	229	2.7	0.6
10° Ext	243	9	0.61
15° Ext	272	22	0.61
0° Add/Abd	121	0	0.45
5° Add	135	11	0.48
10° Add	165	36	0.47
15° Add	240	98	0.42
5° Abd	133	9	0.49
10° Abd	128	5	0.49
15° Abd	125	3	0.49

Note: Flex = Flexion, Ext = Extension, Add = Adduction, Abd = Abduction.

**Table 2 jcm-12-00233-t002:** Patient demographics and results.

	PASTA (SD), *n* = 46	BURSA (SD) *n* = 36	*p*-Value
Male/Female	25/21	17/11	0.64
Mean Age [years]	53 (15.29)	57 (16.02)	0.95
CSA [°]	33.73 (3.67)	34.56 (4.46)	0.062
Acromion Index	0.72 (0.09)	0.73 (0.08)	0.44
Ratio Width to Length of Acromion	0.48 (0.07)	0.44 (0.06)	0.021
Medial Acromial Border Offset [mm]	40.89 (9.20)	45.16 (8.89)	0.021
Short Sclerotic Line [mm]	16.25 (5.02)	13.09 (3.98)	0.008
ACOA [°]	11.76 (4.9)	12.72 (3.75)	0.35
AC Offset [mm]	11.7 (4.94)	12.78 (3.77)	0.29

Note: SD = standard deviation, CSA = critical shoulder angle, ACOA = acromioclavicular offset angle, AC = acromioclavicular.

## Data Availability

Redcap software (see Section 2).

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
