# Peer review of "Acromial Morphology and Its Relation to the Glenoid Is Associated with Different Partial Rotator Cuff Tear Patterns"

_jcm, 2022, doi:10.3390/jcm12010233_

Round 1

Reviewer 1 Report

- Title
It should spesific address the main study findings.

- Abstract

Please describe the extended word of PASTA before using the acronym

- Introduction

After you use the acronym the first time, you should not use the extended word anymore, Line 77 i.e. “articular sided partial tears of the supraspinatus” = PASTA

Please define better the study aim at the end of the introduction.

- Methods
Not clear the exclusion criteria, why were they excluded if they showed acromio-humeral distance of < 7 mm? Isn't this a part of tear development?
How many were exluded?

- Results
Good enough.

- Discussion
Please start with "the main study finding is.."
The central part of the discussion lacks references.
Overall, the discussion is superficial and repeats the study results. It would be important to report the strengths and weaknesses of these studies in other studies in the literature, underlying differences and added value of this study.
Small sample group might be one of limitation of this study

- References
They could be updated and expanded on relevant studies in the field.

- Figures and tables
Good

Author Response

Reviewer 1: Comments and Suggestions for Authors

- Title
It should spesific address the main study findings.

After thorough reevaluation of the title the authors decided to keep " Acromial morphology and its relation to the glenoid is associated with different partial rotator cuff tear patterns", since it describes the study and its findings appropriately.

- Abstract

Please describe the extended word of PASTA before using the acronym

We have addressed this in line 18 and used the original term "partial articular supraspinatus tendon avulsion" (PASTA).

- Introduction

After you use the acronym the first time, you should not use the extended word anymore, Line 77 i.e. “articular sided partial tears of the supraspinatus” = PASTA

We have added the original term "partial articular supraspinatus tendon avulsion" (PASTA) in line 96/97. That is where the term is used for the first time in the manuscript (not the abstract). In the introduction section we have adapted the text slightly for a better readability.

Please define better the study aim at the end of the introduction.

The study aim was added in line 83-85.

- Methods
Not clear the exclusion criteria, why were they excluded if they showed acromio-humeral distance of < 7 mm? Isn't this a part of tear development?
How many were exluded?

Acromio-humeral distance <7mm is associated with full-thickness rotator cuff tears and this was used as one of the exclusion criteria.

We did not exclude any patients in the BURSA or PASTA group who met the inclusion criteria, with a consecutive series of 28 patients in the BURSA and 46 patients in the PASTA group.

M J Scheyerer, FE Brunner, C Gerber, The acromiohumeral distance and the subacromial clearance are correlated to the glenoid version. Orthop Traumatol Surg Res. 2016 May;102(3):305-9.

- Results
Good enough.

- Discussion
Please start with "the main study finding is.."

This was added as requested.

The central part of the discussion lacks references.
Overall, the discussion is superficial and repeats the study results. It would be important to report the strengths and weaknesses of these studies in other studies in the literature, underlying differences and added value of this study.

We have changed parts of the Discussion section and have revised the references. We could identify one further study (by Liu et al, BMC, 2021) that we included in the discussion. However, there is not more data in the literature available regarding omometric parameters for partial rotator cuff tears, which supports the impact of the present study.

Small sample group might be one of limitation of this study

Added as requested.

- References
They could be updated and expanded on relevant studies in the field.

Revised.

- Figures and tables
Good

Reviewer 2 Report

The following points should be addressed before this manuscript can be considered for publication.

·         Abstract: the term “conflict” might be changed to another term, such as interrelation or impact

·         Abstract: The hypothesis should be indicated

·         Methods:  how the final measurements were determined? By agreement between radiologist and surgeon or separately by each of them, and if this is the case, whose interpretation was the final?

·         Results: When no significant difference is reported (p>0.05), the statistical power of the comparison test should be given and discussed; otherwise, it is difficult to interpret the meaning of these results.

Author Response

Reviewer 2: Comments and Suggestions for Authors

The following points should be addressed before this manuscript can be considered for publication.

  • Abstract: the term “conflict” might be changed to another term, such as interrelation or impact

We have changed the term "conflict" to "impact".

  • Abstract: The hypothesis should be indicated

The hypothesis was added to the Abstract. Line 20-22.

  • Methods:  how the final measurements were determined? By agreement between radiologist and surgeon or separately by each of them, and if this is the case, whose interpretation was the final?

 The average of both measurements was used for final analysis. We have added this information to the methods section.

  • Results: When no significant difference is reported (p>0.05), the statistical power of the comparison test should be given and discussed; otherwise, it is difficult to interpret the meaning of these results.

As written in the method section, independent T-tests were used.

We have included all available patients for this study, and as a retrospective study, there are inevitable limitations such as limited power. However, power calculation for this retrospective study was omitted, because post-hoc power analysis is an inappropriate tool for power estimates1,2. Nevertheless, we agree that the sample size is limited, especially with regards to sub-analyses. We mentioned this limitation in the Limitations section.

  1. Hoenig JM, Heisey DM. The abuse of power. The American Statistician. 2001;55(1):19-24.
  2. Mumford JA. A power calculation guide for fMRI studies. Soc Cogn Affect Neurosci. 2012;7(6):738-742.

Round 2

Reviewer 2 Report

The corrections and clarifications are sufficient for publication.